# Subcutaneous Infliximab [CT-P13], a True Biologic 2.0. Real Clinical Practice Multicentre Study

**DOI:** 10.3390/biomedicines10092130

**Published:** 2022-08-30

**Authors:** Jose M. Huguet, Victor García-Lorenzo, Lidia Martí, Jose María Paredes, Jose Joaquin Ramírez, Miguel Pastor, Lucia Ruiz, Ana Sanahuja, Pilar Timoneda, Laura Sanchís, Gloria Alemany Pérez, Marta Maia Boscá-Watts

**Affiliations:** 1Gastroenterology Department, Hospital General Universitario de Valencia, 46014 Valencia, Spain; 2Gastroenterology Department, Hospital Francesc de Borja de Gandia, 46702 Valencia, Spain; 3Gastroenterology Department, Hospital Universitario Doctor Peset, 46017 Valencia, Spain; 4Gastroenterology Department, Hospital Lluis Alcanyis de Xativa, 46800 Valencia, Spain; 5Gastroenterology Department, Hospital Clínico Universitario de Valencia, University of Valencia, 46010 Valencia, Spain; 6Clinical Analysis Department, Hospital General Universitario de Valencia, 46014 Valencia, Spain; 7Gastroenterology Department, Hospital de Sagunto, 46520 Valencia, Spain

**Keywords:** infliximab, CT-P13, infliximab trough level, inflammatory bowel disease, Crohn’s disease, ulcerative colitis, subcutaneous, switch

## Abstract

Background: Inflammatory bowel disease (IBD), including Crohn’s disease and ulcerative colitis, is characterized by chronic relapsing intestinal inflammation. There are few data on the efficacy and safety in clinical practice of infliximab (CT-P13) in subcutaneous formulation (SC) for the treatment of patients with IBD. Methods: Multicenter, prospective study of patients with IBD in clinical remission, who had their treatment changed from intravenous (IV) infliximab to SC. Two groups of patients were evaluated according to whether they were on IV infliximab treatment at standard or intensified doses before the switch. Results: A total of 30 patients were on standard dosing and another 30 in intensified therapy. Treatment persistence in both groups at 6 months was greater than 95%. In both groups after the change, neither the biomarkers of inflammation nor the activity indices underwent significant changes at 3 and 6 months compared to the baseline value. Similarly, in both groups, infliximab trough levels showed a significant increase 3 and 6 months after the change to SC. No serious adverse events were registered. Conclusions: The CT-P13 SC brings a new anti-TNF era. Achieving much higher drug levels that are constant over time opens new paths to explore the management of patients with IBD: less immunogenicity, better perianal disease control and higher achievement of mucosal healing.

## 1. Introduction

Inflammatory bowel disease (IBD) is a heterogeneous group of chronic inflammatory disorders: the main phenotypes comprise ulcerative colitis (UC) and Crohn’s disease (CD). IBD is characterized by chronic relapsing intestinal inflammation. UC is restricted to the colon, whereas CD is characterized by involvement of the gastrointestinal tract from the mouth to anus in a discontinuous fashion, with the development of strictures, abscesses or fistulas that penetrate neighboring organs or the perianal skin. It has been a worldwide healthcare problem with a continually increasing incidence. It is thought that IBD results from an aberrant and continuing immune response to the microbes in the gut, catalyzed by the genetic susceptibility of the individual. Although the etiology of IBD remains largely unknown, it involves a complex interaction between the genetic, environmental or microbial factors and the immune responses [1,2,3].

Since 1998, we have had infliximab in its intravenous (IV) formulation for the treatment of patients with IBD. Infliximab is a monoclonal antibody against tumor necrosis factor alpha (TNFα) [4]. In Europe, we have two other anti-TNFα drugs such as adalimumab [5] and golimumab [6], both of which are available in subcutaneous (SC) formulation. CT-P13, a biosimilar of intravenous infliximab, was developed in 2013, and has been available in our daily practice since 2016 [7]. To date, data from real-life cohorts and randomized controlled trials show comparable clinical efficacy, safety, and immunogenicity of biosimilar CT-P13 and the original drug [8]. In addition, in real clinical practice, it has been shown that the switch from the original drug Remicade^®^ to the biosimilar CT-P13 is safe and effective [9].

A randomized trial has recently been published in which the efficacy and safety of a subcutaneous formulation of infliximab (CT-P13) [10] was evaluated. This study compared pharmacokinetics, symptomatic and endoscopic efficacy, safety and immunogenicity of a subcutaneous formulation of the infliximab biosimilar CT-P13 SC vs. intravenous CT-P13 in patients with IBD. This study concluded that the pharmacokinetic non-inferiority of SC CT-P13 versus IV CT-P13 and comparable efficacy, safety and immunogenicity profiles supported the potential suitability of SC CTP13 treatment in IBD. In this same study, it was observed that in those patients who had been randomized to intravenous CT-P13, when switched to subcutaneous treatment at week 30, drug levels rose to equal the highest values achieved with the arm randomized to subcutaneous CT-P13 from baseline. Likewise, no differences were found between groups after week 30 concerning clinical activity, biomarkers, mucosal healing or quality of life. However, we need evidence from clinical practice to reaffirm these results. To date, only one study has been published in manuscript form that evaluated the persistence of treatment, as well as the pharmacokinetic data of subcutaneous infliximab [11]. The manuscript by Smith et al. is a retrospective multicenter cohort study, which included stable patients who received treatment with IV infliximab and who were switched to SC CT-P13. In this study, the authors conclude that this change presents high rates of treatment persistence and low rates of immunogenicity. In addition, no changes were observed in disease clinical activity indices or in biomarkers. As in the pivotal study, infliximab levels increased after switching to SC CT-P13, and only the presence of antibodies was associated with serum infliximab levels [11].

On the other hand, it has been observed that there is a positive correlation between the concentrations of biological drugs and favorable therapeutic results in IBD, which has mainly been shown with infliximab [12].

## 2. Materials and Methods

### 2.1. Study Design

This is a prospective, multicenter study of patients with IBD who were receiving intravenous infliximab treatment in 6 hospitals in Valencia (Spain). Data were collected based on the clinical practice of the participating physicians; the specific therapeutic strategy was not decided in advance by the study protocol but was determined by the usual practice of medicine, and the decision to prescribe or not the switch to subcutaneous CT-P13 was dissociated from the decision to include the patient in the study.

### 2.2. Patients

All those patients who, due to clinical practice, were offered a change of treatment from intravenous infliximab (Remicade^®^, Remsima^®^ or Inflectra^®^) to subcutaneous CT-P13 (Remsima^®^) were eligible. For the switch, patients had to be in clinical remission for at least 3 months defined as a Harvey-Bradshaw index < 4 (HBI) for CD [13] or a partial Mayo index < 2 (PMI) for UC [14], and without taking corticosteroids in the previous 3 months. Two groups of patients were evaluated: the standard regimen group (dosage of 5 mg/kg every 8 weeks) and the intensified group (any dosage greater than 5 mg/kg every 8 weeks). All patients were switched to a standard SC dose of CT-P13 of 120 mg every two weeks. Patients who had not completed at least 3 months of follow-up at the time of analysis were excluded.

### 2.3. Data Collected

We collected baseline clinical information that included age, sex, weight, disease progression time, disease extent according to the Montreal classification, the reason for infliximab treatment initiation, infliximab treatment duration time, dosage of intravenous infliximab, previous biological treatments, concomitant therapy with immunomodulators or steroids, activity indices (HBI and PMI) and the brand of intravenous infliximab. Likewise, infliximab levels were collected on the day of the change to CTP-13 and C-reactive protein (CRP) and fecal calprotectin (FCP) no later than the previous 15 days. At follow-up, data were collected at months 3, 6, 9 and 12 after the switch. Follow-up data included HBI, PMI, CRP, FCP and serum infliximab levels (as well as the presence of anti-drug antibodies). Data were included if collected within 15 days before assessment at each specified time point. However, all infliximab drug levels were collected within 24 h before the next SC CT-P13 injection or immediately before this dose. Adverse events and their possible relationship with treatment were also collected, as well as the need for surgery or hospitalization during follow-up. Discontinuation of treatment and the reason for it (loss of efficacy, adverse event, complication requiring surgery, physician’s clinical decision or patient decision), as well as loss of follow-up, were recorded.

We present data at 6 months of follow-up.

### 2.4. Determination of Drug Levels and Anti-Drug Antibodies

In our study, drug and antidrug serum levels were measured using two different kits. One of them was the LISA-TRACKER kit (Theradiag). This is an enzyme-linked immunosorbent assays (ELISA) for the quantitative determination of Infliximab and anti-Infliximab antibodies. Regarding the levels of Infliximab detected, the assay range covers all clinically relevant drug concentrations (Infliximab: 0.3–20 μg/mL). Regarding the determination of the antibodies by the Lisa-Tracker Kits, the assay range is Antiinfliximab: 10–200 ng/mL. In one of the centers the kits used were Progenika kits [Promonitor-IFX, Promonitor-anti-IFX], it is an ELISA and its assay range is 0.035–14.4 μg/mL. In recent months two centers have changed the measurement system to i-TrackerInfliximab (Theradiag) which is an automated chemiluminescence immunoassay on human serum or plasma samples. Their assay ranges are 0.3–24 µg/mL.

### 2.5. Study Objectives

The aim of our study was to evaluate the persistence of SC CT-P13 in patients with IBD who were switched from intravenous infliximab to subcutaneous administration. The secondary aims were the following: to determine the levels of infliximab before and in each of the three-monthly evaluations, to assess the possible clinical changes (using the HBI for CD and the PMI for UC), to evaluate the possible changes in the inflammatory biomarkers (CRP and FCP) and determine the occurrence of possible side effects attributable to the drug.

### 2.6. Statistical Analysis

We used basic descriptive statistics, which included the mean and standard deviation for continuous variables and absolute frequency and percentages for discrete variables. To determine de differences between clinical indices, serum and fecal biomarkers and serum infliximab concentrations; the Friedman test was used. The Statistical Package for the Social Sciences (SPSS) 22 was used to describe and analyze the data, considering *p* values < 0.05.

### 2.7. Ethics

The study was conducted in accordance with the Declaration of Helsinki and approved by the Ethics Committee of Consorcio Hospital General Universitario de Valencia on 8 April 2021, and the reference number is 142/2021. Informed consent was obtained from all subjects involved in the study.

## 3. Results

### 3.1. Study Population

Of the 77 patients to whom the change was proposed, 60 [78%] accepted the change to SC IFX. Of these 32 [53.3%] were male and had Crohn’s disease 33 [55%]. A total of 49 patients switched from CT-P13 (Remsima^®^ or Inflectra^®^) and 11 from original infliximab (Remicade^®^). A total of 30 patients were on standard dosing and another 30 in intensified therapy. In Table 1 we can see the baseline characteristics of the patients according to the previous dosage (standard or intensified). The different intensifications were: 5 mg/kg every 7 weeks (1 patient, 3.3%), 5 mg/kg every 6 weeks (15 patients, 50%), 5 mg/kg every 5 weeks (1 patient, 3.3%) and 5 mg/kg every 4 weeks (13 patients, 43.4%).

The main indication for starting treatment with IV infliximab had been failure of previous conventional treatment in 53.3% of cases, the second being corticoresistance in 18.3%. Only in 1 patient [1.7%] the initial indication had been the presence of active perianal disease.

### 3.2. Treatment Persistence and Clinical Indices

#### 3.2.1. Standard Regimen Patients

In the short term (3 months), all patients maintained treatment. At the time of the analysis, 21 patients had completed 6 months of follow-up, in which the persistence of treatment was 95.2% (only one patient did not continue in treatment due to his decision not to continue. He was in clinical remission).

The clinical indices remained at remission values (HBI < 5 and PMI < 2) except for one patient who presented a mild flare-up at 3 months and was treated with budesonide.

#### 3.2.2. Intensified Patients

In the short term (3 months), all patients persisted with treatment and also remained in remission based on clinical indices. Of the initial 30 patients, at the time of the analysis, 17 had already completed the 6-month follow-up and all of them were still on treatment and in clinical remission. One patient had an HBI of 6 that was attributed to previous surgery for his CD.

### 3.3. Biomarkers, Pharmacokinetics and Anti-Infliximab Antibodies

#### 3.3.1. Standard Regimen Patients

Table 2 shows how the biomarkers of inflammation (CRP and FCF) did not undergo significant changes at 3 and 6 months compared to the baseline value. We also observe that the trough levels of infliximab present a significant increase (Table 2 and Figure 1) at 3 and 6 months after the change to SC.

Anti-IFX antibodies were detected in only one patient with very low IFX levels, that were already present before the switch. At three months, one patient had withdrawn azathioprine without observing clinical changes, and this withdrawal was maintained at 6 months of follow-up.

#### 3.3.2. Intensified Patients

Table 3 shows how the biomarkers of inflammation (CRP and FCF) did not undergo significant changes at 3 and 6 months compared to the baseline value, and the trough levels of infliximab suffered a significant increase (Table 3 and Figure 2) at 3 and 6 months after the change to SC.

No patient had anti-drug antibodies. At three months of follow-up, three patients had withdrawn azathioprine and the patient on methotrexate was also able to remove it, without observing clinical changes. These withdrawals were maintained at 6-month follow-up without observing any clinical worsening. 

### 3.4. Intensifications or Switched Back to IV Infliximab after Switch to Subcutaneous Formulation

No patient required an increase in the standard dosage of 120 mg/every 2 weeks nor did any patient return to intravenous dosing either due to adverse events or clinical decision.

### 3.5. Outcomes of Perianal CD

A total of nine patients had perianal disease (five in the standard dosage group and four in the intensified group). The months of treatment with IV infliximab prior to the change were a median of 30 months (interquartile range (IQR) 73). No clinical worsening of perianal disease was observed in any of them after switching to SC. On Table 4, we can see the variation in the levels of CT-P13 after the change to SC, with an upward trend.

### 3.6. Adverse Events

Regarding the adverse events described, only one patient had pain and erythema at the injection site that did not require withdrawal or change. Another patient presented mild elevation of transaminases (<5×) that she had already presented on other occasions and that was not considered to be related to the drug.

No patient required neither hospital admission nor surgery for their IBD.

## 4. Discussion

Our study shows that SC CT-P13 maintains high rates of drug persistence beyond 6 months. Both the HBI and the PMI remained at inactive values throughout the follow-up, as did the biomarkers (CRP and FCP). Infliximab levels increased significantly with the switch to subcutaneous CT-P13.

After more than 20 years of existence of infliximab in its intravenous formulation, the appearance of the subcutaneous formulation represents a revolution in the management of the drug and places it as a true “infliximab 2.0”. We think that this is due to its pharmacokinetic advantages that have been evaluated both in clinical trials and in real clinical practice data, such as those shown in the present study. We think that this is due to its pharmacokinetic advantages that have been evaluated both in clinical trials and in real clinical practice data, such as those shown in the present study.

In our study, the rate of drug persistence was high both at 3 and 6 months, in agreement with what was previously published by Smith et al. [11] in which one year after the switch the persistence was 92.3%.

As in other studies published to date [10,11,15,16,17], both in intensified patients and those with a standard regimen, infliximab trough levels increased significantly. They rose from 12.7 μg/dL at baseline to 18.9 μg/dL in the intensified ones and from 5.2 μg/dL to 17.6 μg/dL in the non-intensified ones. This may have its significance in relation to immunogenicity and, therefore, have influenced the persistence of the drug. In our study, we did not have any patient with intensifications greater than that indicated in the data sheet [18] (namely 10 mg/kg of infliximab every 8 weeks or 5 mg/kg every 4 weeks); however, in the study by Buisson et al. [15], they observed that in patients with intensifications greater than what is indicated in the data sheet, there was no increase in the trough levels of infliximab compared to baseline with the switch to subcutaneous. In our patients, drug levels were determined within 24 h prior to the following SC CT-P13 injection or immediately before this dose. However, it has been described that drug levels would remain stable throughout the 14 days of the treatment cycle, so the determination would not be subject to trough levels as occurs with intravenous dosing [19].

It has been suggested that patients with CD and perianal involvement would require higher drug levels to achieve improvement of the fistulous disease [20,21]. In our study, we observed that this subgroup of patients also achieved a significant increase in plasma levels of infliximab (pre-change 7.6 μg/dL vs. 20.27 μg/dL at 6 m) without observing clinical worsening (evaluated for clinical symptoms and pus discharge after digital pressure). These data are similar to those reported by Smith et al. [11] in which, in addition to the significant increase in plasma levels of infliximab, only 8% of patients experienced clinical worsening. It has also been suggested that those patients with a severe flare of ulcerative colitis would require higher levels of drug to avoid colectomy, due to a higher clearance rate of infliximab [22,23,24]. In our study, no patient was in this situation, because the inclusion criterion was to be in clinical remission. However, this is a path to be explored in future studies, again based on the drug levels achieved in patients with ulcerative colitis. Higher levels of infliximab have also been shown to correlate with greater mucosal healing [25], with Ungaro et al. proposing serum infliximab levels of 6 to 10 μg/mL to achieve mucosal healing in 80% to 90% of patients with IBD, and that this could be considered a “therapeutic window” [26]. In our study, endoscopy was not systematically performed on the patients before and after the change, so this aspect could not be analyzed.

The rate of acceptance of the change was 78%. These are results similar to those reported by Buisson et al. [15], in which 71.8% of the patients accepted the change and higher than those reported by Cummings S et al. [27] in who acceptance was 59%.

Drug tolerance was excellent. Regarding the adverse events described, they were mild and did not require drug withdrawal, in accordance with what has been published to date [10,11].

The association of immunosuppressive and anti-TNF drugs has been associated with an increased risk of infections and other adverse events [28,29]. On the other hand, it has been postulated that patients treated with anti-TNF and carriers of the HLADQA1*05 allele would have higher immunogenicity rates, and therefore, would benefit from the association of immunosuppressive drugs with anti-TNF [30]. In our registry, we did not withdraw immunosuppressants by default from patients who were taking them at the start of the registry (this only happened in five patients). However, in view of the high drug levels achieved, we consider that this is a field to be explored with the consequent benefit that its withdrawal could have in terms of minimizing related adverse events.

Once again, in view of the results in drug levels, we consider that another field to explore is the spacing of treatment beyond 2 weeks in patients with very high levels and in clinical-biological remission. In our group, and after 9 months of follow-up, this possibility has been explored in 2 patients in whom we spaced treatment for 3 weeks, having observed maintenance of clinical remission 3 months after spacing.

Several methodologies have been developed to measure serum concentrations of anti-TNFα drugs, including different types of ELISA, radioimmunoassay, liquid chromatography-based homogeneous mobility shift assay, liquid chromatography coupled to mass spectrometry, etc. [31]. Among them, ELISA-based kits are the most widely used in clinical laboratories. The ELISA tests used in this work have shown excellent correlation with other commercially available assays used for drug monitoring [32,33]. Likewise, no differences have been observed in the determination of the original infliximab or the biosimilar, nor in its interpretation [34].

## 5. Conclusions

In conclusion, the subcutaneous infliximab formulation, CT-P13, brings a new anti-TNF era. Achieving much higher drug levels that are constant over time opens new paths to explore the management of patients with IBD: less immunogenicity with the possibility of not combining immunosuppressant drugs, better perianal disease control, higher achievement of mucosal healing, the possibility of using it in patients with severe ulcerative colitis with high inflammatory load and even spacing treatments beyond 2 weeks.

## Figures and Tables

**Figure 1 biomedicines-10-02130-f001:**
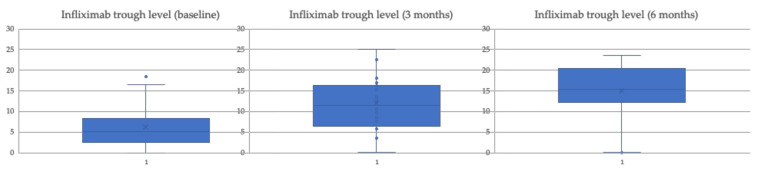
Trends in serum infliximab concentrations (μg/dL) standard dose patients.

**Figure 2 biomedicines-10-02130-f002:**
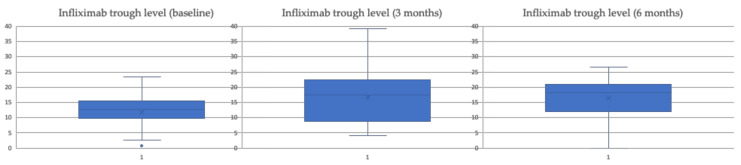
Trends in serum infliximab concentrations [μg/dl] intensified patients.

**Table 1 biomedicines-10-02130-t001:** Baseline characteristics of patients switched to SC CT-P13.

	All [*n* = 60]	Standard Dosage [*n* = 30]	Intensified [*n* = 30]
Age, mean [SD]	39.3 [13.4]	37,7 [12.3]	41.9 [14.5]
Sex, men, n [%]	32 [53.3%]	15 [50%]	17 [56.7%]
Weight [kg], mean [SD]	72.2 [15.03]	72.47 [16.14]	71.98 [14.28]
Disease type			
CD, n [%]	33 [55%]	16 [53.3%]	17 [56.7%]
UC, n [%]	27 [45%]	14 [46.7%]	13 [43.3%]
Montreal classification:			
Age, n [%]			
A1 [<16]	5 [15.1%]		2 [11.7%]
A2 [17.0–40.0]	23 [69.7%]	3 [18.8%]	12 [70.5%]
A3 [>40]	2 [6.1%]	11 [68.8%]	0
ND	3 [9.1%]	2 [12.5%]	3 [17.6%]
Disease extent, n [%]			
Ileal [L1]	16 [48.5%]		9 [53%]
Colonic [L2]	6 [18.2%]	7 [43.8%]	0
Ileo-colonic [L3]	8 [24.2%]	6 [37.5%]	5 [29.4%]
Upper GI [L4]	0	3 [18.8%]	0
ND	3 [9.1%]		3 [17.6%]
Behaviour classification, n [%]			
Non-S-Non-P [B1]	21 [63.7%]	12 [75%]	9 [53%]
Stricturing [B2]	4 [12.1%]	2 [12.5%]	2 [11.7%]
Penetrating [B3]	5 [15.1%]	2 [12.5%]	3 [17.6%]
ND	3 [9.1%]		3 [17.6%]
Perianal disease, n [%]	9 [27.3%]	5 [31.3%]	4 [23.5%]
UC Montreal extent, n [%]			
Proctitis [E1]	1 [3.7%]	0	1 [7.7]
Left-side colitis [E2]	11 [40.7%]	4 [28.6%]	7 [53.8%]
Pancolitis [E3]	15 [55.6%]	10 [71.4%]	5 [38.5%]
Months from diagnosis,			
mean [SD]	10 [7.5]	8.5 [5.2]	11.5 [9.2]
Months with IFX treatment,			
median [IQR]	22 [46.0]	34 [43.75]	23 [62.0]
Previous anti-TNF, yes, n [%]	11 [18.3%]	6 [20%]	5 [16.6%]
Other previous biologics:			
Vedolizumab, n [%]	1 [1.6%]		
Ustekinumab, n [%]	1 [1.6%]		1 [3.3%]
Ciclosporine, n [%]	1 [1.6%]	1 [3.3%]	1 [3.3%]
Concomitant			
immunomodulator, n [%]			
Thiopurines	29 [48.3%]	14 [46.7%]	15 [50%]
Metotrexate	2 [3.3%]	1 [3.3%]	1 [3.3%]
Esteroids baseline, n [%]	0	0	0
HBI, median [IQR]	0 [2]	0 [1]	0.5 [2]
PMI, median [IQR]	0 [0.75]	0 [0.25]	0 [1.25]
CRP, mg/dL^a^, median [IQR]	0.3 [0.62]	0.25 [0.3]	0.4 [0.73]
FCP, mcg/g^b^, median [IQR]	53 [117.0]	30 [64.0]	93 [216.0]
Infliximab trough level μg/dL, median, [IQR]	8.4 [9.77]	5.2 [5.8]	12.7 [5.8]

SD = standard deviation; kg = kilogram; CD = Crohn’s Disease; UC = ulcerative colitis; ND = not available; Non-S-Non-P = non-stricturing, non-penetrating; IFX = infliximab; anti-TNF = anti tumoral necrosis factor; IQR = interquartile range; HBI = Harvey-Bradsaw index; PMI = Partial Mayo Index; CRP = C-reactive protein; FCP = fecal calprotectin; a = normal CRP value < 5 mg/L; b = normal FCP value < 200 µg/g.

**Table 2 biomedicines-10-02130-t002:** Trends in biomarkers and infliximab trough levels in standard regimen patients.

Variable	Baseline	3 Months	*p*-Value	6 Months	*p*-Value
CRP, mg/dL ^a^, median [IQR]	0.25 [0.3]	0.3 [0.5]	*p* > 0.05	0.2 [0.27]	*p* > 0.05
FCP, mcg/g ^b^, median [IQR]	30 [64.0]	36 [98.5]	*p* > 0.05	24 [51.0]	*p* > 0.05
IFL μg/dL, median, [IQR]	5.2 [5.8]	11.4 [9.86]	*p* = 0.001	17.6 [8.4]	*p* < 0.001

CRP = C-reactive protein; FCP = fecal calprotectin; IFL = Infliximab trough level; IQR = interquartile range; ^a^ = normal CRP value <5 mg/L; ^b^ = normal FCP value < 200 µg/g.

**Table 3 biomedicines-10-02130-t003:** Trends in biomarkers and infliximab trough levels in intensified patients.

Variable	Baseline	3 Months	*p*-Value	6 Months	*p*-Value
CRP, mg/dL ^a^, median [IQR]	0.4 (0.73)	0.3 (0.5)	*p* > 0.05	0.4 (0.57)	*p* > 0.05
FCP, mcg/g ^b^, median [IQR]	93 (216)	62 (128)	*p* > 0.05	36 (367)	*p* > 0.05
IFL μg/dL, median, [IQR]	12.7 (5.8)	17.4 (13.53)	*p* = 0.007	18.9 (7.48)	*p* = 0.004

CRP = C-reactive protein; FCP = fecal calprotectin; IFL = Infliximab trough level; IQR = interquartile range; ^a^ = normal CRP value <5 mg/L; ^b^ = normal FCP value < 200 µg/g.

**Table 4 biomedicines-10-02130-t004:** Trends in infliximab trough levels in patients with perianal CD.

Variable	Baseline	3 Months	*p*-Value	6 Months	*p*-Value
ITL μg/dL, mean (SD)	7.6 [6.4]	13.95 [6.0]	*p* = 0.014	20.27 [2.9]	*p* = 0.083

CD = Crohn’s disease; IFL = Infliximab trough level; SD = standard deviation.

## Data Availability

All data generated or analyzed during this study are included in this article. Further enquiries can be directed to the corresponding author.

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
