# Peer review of "Subcutaneous Infliximab [CT-P13], a True Biologic 2.0. Real Clinical Practice Multicentre Study"

_biomedicines, 2022, doi:10.3390/biomedicines10092130_

Round 1

Reviewer 1 Report

Huguet et al. submitted the manuscript, "Subcutaneous infliximab [CT-P13], a true biologic 2.0. Real clinical practice multicentre study" is about the clinical study where patients with inflammatory bowel disease (IBD) were treated with Subcutaneous infliximab [CT-P13].

The main strength of the paper is the inclusion of patients from a broad background; out of 77, 53% were male, and 55% had Crohn's disease.

Although the paper provides a clinical evaluation of infliximab [CT-P13] and is well organized, there are a few minor points that need to be addressed.

1.              The authors emphasized on previously reported studies, where only references 7 and 8 were in focus. If authors include a small table covering the important points/observations from these studies, It would certainly improve the readability of the current paper.

2.              There are typographical errors.

Please go through the paper and revise them, such as

(a) Page 2, line 55, there should be a full stop before the sentence starts, "In the English………………."

(b) There is some typographical mistake in the authors' affiliations. Please revise it accordingly to the journal guidelines.

3.               Choice of words can be improved, such as mentioned above, "In the English study……...." can be replaced with a better choice of words. For example, in my suggestion, "First author surname et al.

The paper is written systematically and has key elements to qualify for the current journal.

Reviewer 2 Report

The study is well organized and the manuscript is well writte, The authors know very well the clinic and the effect of infliximab. Reading the paper is probably missed a part dedicated to the disease and eventual description. This is more important also for eventual clinician that don't know the IBD. May the authors in introduction add to this point?
